

# The responses of $CO_2$ emission to nitrogen application and earthworm addition in the soybean cropland

Mei Guang Jiang[1], Jingyuan Yang[1], Qi Xu[1], Linyu Qi[1], Yue Gao[1], Cancan Zhao[1,2], Huijie Lu[1], Yuan Miao[1,2] and Shijie Han[1]

[1] School of Life Sciences, Henan University, Henan, China
[2] Henan Dabieshan National Field Observation and Research Station of Forest Ecosystem, Xinyang Academy of Ecological Research, Xinyang, China

## ABSTRACT

The effects of nitrogen application or earthworms on soil respiration in the Huang-Huai-Hai Plain of China have received increasing attention. However, the response of soil carbon dioxide ($CO_2$) emission to nitrogen application and earthworm addition is still unclear. A field experiment with nitrogen application frequency and earthworm addition was conducted in the Huang-Huai-Hai Plain. Results showed nitrogen application frequency had a significant effect on soil respiration, but neither earthworms nor their interaction with nitrogen application frequency were significant. Low-frequency nitrogen application (NL) significantly increased soil respiration by 25%, while high-frequency nitrogen application (NH), earthworm addition (E), earthworm and high-frequency nitrogen application (E*NH), and earthworm and low-frequency nitrogen application (E*NL) also increased soil respiration by 21%, 21%, 12%, and 11%, respectively. The main reason for the rise in soil respiration was alterations in the bacterial richness and keystone taxa (Myxococcales). The NH resulted in higher soil nitrogen levels compared to NL, but NL had the highest bacterial richness. The abundance of Corynebacteriales and Gammaproteobacteria were positively connected with the $CO_2$ emissions, while Myxococcales, Thermoleophilia, and Verrucomicrobia were negatively correlated. Our findings indicate the ecological importance of bacterial communities in regulating the carbon cycle in the Huang-Huai-Hai Plain.

## INTRODUCTION

Large amounts of $CO_2$ are released into the environment by soil through respiration, which raises atmospheric $CO_2$ concentrations and threatens ecological sustainability (*Bond-Lamberty & Thomson, 2010*). The carbon budget of terrestrial ecosystems is affected by even minor changes in soil respiration (*Heimann & Reichstein, 2008*). As one of the most active parts of the terrestrial ecosystem, the agricultural system is crucial to the global carbon cycle (*Crippa et al., 2021*). Soil carbon emissions from cropland should be understood to predict and manage soil carbon stores (*Wu et al., 2019*). The main components of soil respiration are microbial and plant root respiration, both of which are

Corresponding authors
Yuan Miao,
miaoyuan0921@126.com
Shijie Han, hansj@iae.ac.cn

regulated by biotic and abiotic factors (*Chen et al., 2019*; *Lei et al., 2021*), such as root dynamics, microclimate, substrate availability, nutrition levels, and soil microbial activity (*Allison, Wallenstein & Bradford, 2010*; *Talmon, Sternberg & Grünzweig, 2011*; *Wagai et al., 2013*; *Bolat & Ozturk, 2017*; *Wang et al., 2019b*). These variables have been incorporated into parameterizing models that forecast shifts in the global carbon cycle. However, a large proportion of the variation in soil respiration remains unexplained, which limits our capacity to forecast carbon cycling under scenarios of climate change (*Liu et al., 2020b*). Microbial ecology is one of the most promising fields in the hunt for novel indicators of soil carbon cycling (*Liu et al., 2020b*). Soil microorganisms play significant roles in predicting $CO_2$ emission through microbial processes (*Liu et al., 2018*). Both theoretical and practical evidence point to the possibility of predicting soil carbon fluxes using the functional and taxonomic characteristics of soil microbial communities (*Liu et al., 2021*; *Allison, Wallenstein & Bradford, 2010*; *Liu et al., 2018*). For instance, changes in the soil microbial community composition have an impact on soil carbon respiration and fixation (*Monteux et al., 2018*; *Müller et al., 2018*). Previous studies have revealed that the composition of these microbial communities may play a role in regulating $CO_2$ emissions (*Chen et al., 2021b*; *Wang et al., 2019a*) since copiotrophs have a faster respiration rate than oligotrophs and that Proteobacteria and Actinobacteria are positively connected with $CO_2$ emissions (*Liu et al., 2018*; *Chen et al., 2021b*; *Liu et al., 2020b*). A better understanding of how changes in the structure and composition of microbial communities affect $CO_2$ emissions in agricultural systems is critical to accurately predict climate change-terrestrial carbon cycling feedbacks.

Fertilization is typically thought to be the primary method for increasing crop yield, it also has a significant impact on the carbon pool and carbon flux in the soil. The structure and function of the world's ecosystems are significantly impacted by changes caused by anthropogenic nitrogen application (*Cao et al., 2021*). The influence of nitrogen application is still being debated despite many nitrogen addition experiments being carried out to examine how ecosystem carbon exchange mechanisms respond to nitrogen application (*Cao et al., 2019*; *Yang et al., 2020*). The frequency of nitrogen amendment is a key factor for simulating nitrogen application. The frequency of nitrogen addition has many effects on the ecosystem. For example, under different nitrogen addition frequencies, nitrogen accumulation, plant nitrogen concentration, plant species loss, and plant biomass are significantly different (*Ning et al., 2022*). However, the effects of nitrogen application frequency on soil respiration in soybean cropland is still unclear.

In soil formation and function processes, earthworms play a crucial role as keystone detritivores and ecosystem engineers (*Fonte, Hsieh & Mueller, 2023*; *Yang et al., 2019*; *Fahey et al., 2013*). They can affect soil carbon dynamics since they are ecosystem engineers living in the soil (*Jennings & Watmough, 2016*). Previous research has demonstrated that soil $CO_2$ emissions can increase due to earthworm invasion (*Lubbers et al., 2013*). Through their interactions with microbes, macro, and microfauna, earthworms greatly influence the decomposition process and increase heterotrophic activity, which in turn affects soil carbon dioxide emissions (*Fisk et al., 2004*). Earthworms directly or indirectly affect nitrogen cycle and have the potential to alter ecosystem

functions and services in relation to nitrogen cycle (*Xue et al., 2022*). Nevertheless, there is still limited research on the effects of nitrogen application frequency and earthworm addition on ecosystem carbon emissions. Therefore, disentangling how nitrogen application frequency and earthworm addition affect $CO_2$ emission and its relationship with the soil microbial community is of great significance for mediating C cycling in farmland.

In this study, a field experiment with six treatments was performed in the Huang-Huai-Hai Plain. The specific questions of this study we tried to address are: (1) How does $CO_2$ emission respond to nitrogen application frequency and earthworms? (2) What were the underlying mechanisms influencing $CO_2$ emission response to nitrogen application frequency and earthworms?

# MATERIALS AND METHODS

## Study site

This research was performed at the farm of Jinming Campus of Henan University, Kaifeng City, Henan Province, China (34°49′N, 114°18′E). The crop was soybean and the variety was Kaidou 1104. A permanent $25 \times 10$ m$^2$ rainout shelter with steel frames and covered with a clear polyethylene roof was built in late summer 2021 to control precipitation inputs each year, to avoid the death of soybean and earthworms caused by extreme rainfall. The appropriate rainfall amounts were selected to simulate natural precipitation in the local area with the long-term rainfall means. The region belongs to the temperate continental monsoon climate with 14 °C of annual mean temperature and 650 mm (80% occurring between July and August) of annual average precipitation. The soil texture is sandy loam.

## Experimental design

The experiment used a randomized block design involving two factors of nitrogen and earthworm, including six treatments: control (C), earthworm addition (E), high-frequency nitrogen application (NH), low-frequency nitrogen application (NL), earthworm and high-frequency nitrogen application (E*NH), and earthworm and low-frequency nitrogen application (E*NL). Each treatment was replicated five times with an area of 1 m × 1 m per plot. The total gram of earthworms (*Metaphire guillelmi*) was controlled at 8.0–8.9 g/m$^2$ (about 2–4 earthworms/m$^2$) (*Li et al., 2022*). The endophytic earthworms (*Metaphire guillelmi*) used in the experiment were purchased from breeding company. Each plot was surrounded by glass to prevent earthworms from escaping. The glass fits tightly into the plot and has enough height (40 cm) to prevent earthworms from escaping. Nitrogen (urea) was added by water dissolving and root topdressing. The total amount of nitrogen added was the same as the conventional local field nitrogen application. The experiment included two frequencies of N application (two times *vs.* 12 times): High frequency nitrogen was applied once every seven days from July 15th to September 30th, and low frequency nitrogen was applied once every 30 days from July 15th to September 30th. From seedling stage (VE) to drum stage (R6), high-frequency nitrogen was uniformly added 12 times with 29 N kg·hm$^{-2}$ each time and low-frequency nitrogen was uniformly added twice with

174 N kg·hm$^{-2}$ each time. The first addition of earthworms was made at the soybean seedling stage in July 15th (*Houida et al., 2022*), and we continuously monitored the biomass of earthworms by electroshocking, and a second addition was made at the soybean podding stage in August 21st of 2022 to maintain a constant biomass (*Wang, 2022*).

## Measurement of soil respiration

Soil respiration and temperature were measured every seven days during the soybean growing season using a Li-8100 portable soil $CO_2$ flux system (Li-Cor, Inc., Lincoln, NE, USA) and a thermocouple probe (Li-8100-201; Li-Cor, Inc., Lincoln, NE, USA) connected to the Li-8100 from June 2022 to October 2022. Soil volumetric water content at 0–10 cm soil depth was determined adjacent to each collar using a soil detector (TR-6D).
All measurements were performed between 9 a.m. and 11:30 a.m. To avoid the respiration of aboveground parts of plants and litter decomposition, all living plants and litter inside the collars were removed by hand 2 days before soil respiration was measured. If it rains heavily, our measurement would be postponed for 2 days.

## Soil sampling and analysis and plant index measurements

In September 2022. Three soil cores were randomly collected from each plot at a depth of 0–10 cm using a soil corer (inner diameter 5 cm) and mixed into one, sieved through a 2 mm mesh to separate gravel and roots, and divided into three parts. One subsample was stored at 4 °C for the analysis of the chemical properties of soil *i.e.*, the available ammonium and nitrate using a colourimetric method (Smart Chem 200 Discrete Auto Analyser; Systea, Anagni, Italy). Another subsample was air-dried and ground for analysis of pH, total N (TN) and total carbon (TC). The soil pH was measured with a soil pH meter (TR-6D). The soil TN and TC concentrations were measured by a Vario ELIII Elementar (Elementar Analysensysteme GmbH, Langenselbold, Germany) elemental analyzer.
The third part was stored at −20 °C for the analysis of the microbial community diversity composition spectrum (*Wang et al., 2022*).

After removing the whole plant from the soil, rinsed it slowly with running water to separate the above and below-ground parts of the plant from the cotyledon nodes.
The washed roots were dried in an oven at 65 °C to a constant weight, and the root biomass (RB) was weighed. Aboveground biomass (AGB) was weighed after two weeks of natural air drying. Grain yield (GRY) was measured by removing the mature pods from the plants, placing them in paper bags, and leaving them in a ventilated place for drying to constant weight. The plant height (PLH) was measured by selecting three plants from each plot and measuring them with a tape measure. The number of pods per plant (NPP) and number of grains per plant (NGP) were also artificially measured by choosing three plants from each plot (*Ji et al., 2017*). A total of 100 grain weight (W100) was chosen at random from the grain yield of each plot and weighed using a precision scale (*Ji et al., 2017*).

## DNA extraction, PCR amplification, and Illumina sequencing

Data were collected as previously described in *Li et al. (2022)*. Specifically using E.Z.N.A Soil DNA Kit (Omega Bio-Tek, Norcross, GE, USA), soil DNA was extracted from each

sample in accordance with the manufacturer's protocol. The purity and concentration of the extracted DNA were determined by a NanoDrop-2000 spectrophotometer (Thermo Fisher Scientific, Waltham, MA, USA) (*Li et al., 2022*). Purified soil DNA was fully pooled together after quantitative determination and then for downstream manipulations (*Wang et al., 2022*). The V3–V4 of bacterial 16S rRNA genes were amplified with the following universal primer set: upstream primers 338F (5′-ACTCCTACGGGAGGCAGCA-3′) and downstream primers 806R (5′-GGACTA CHVGGGTWTCTAAT3′). For fungi, the primers ITS5 (5′-GGAAGTAAA AGTCGTAACAAGG -3′) and ITS2 (5′-GCTGCG TTC TTCATCGATGC-3′) (*Usyk et al., 2017*) were used to amplify the ITS_V1 region of the rDNA gene. PCR reactions were performed in 25 µL reaction mixtures containing 5 µL 5 × reaction buffer, 5 µL 5 × GC buffer, 2 µL 2.5 mM dNTPs, 1 µL Forwardprimer (10 uM), 1 µL Reverseprimer (10 uM), 2 µL DNA Template, 8.75 µL ddH$_2$O, 0.25 µL Q5 DNA Polymerase. The reaction conditions were programmed of an initial denaturing step at 98 °C for 2 min, denaturation 98 °C 15 s, annealing 55 °C 30 s, extension 72 °C 30 s, final extension 72 °C 5 min and 10 °C hold 25–30 cycles. Samples were sequenced in an Illumina MiSeq High-Throughput Sequencing (HTS) platform (Illumina, San Diego, CA, USA) at Personal Biotechnology Co. Ltd Shanghai, China to determine soil microbial community composition.

## Statistical analyses

Data were collected as previously described in *Wang et al. (2022)*. Specifically, the sequenced data was performed using QIIME 2 2019.4 with slight modification. Raw sequence data were demultiplexed using the demux plugin followed by primers cutting with cutadapt plugin. Sequences were then merged, filtered and dereplicated using functions of fastq_mergepairs, fastq_filter, and derep_fulllength in Vsearch. All the unique sequences were then clustered at 98% (*via* cluster_size) followed by chimera removing. At last, the non-chimera sequences were re-clustered at 97% to generate OTU representative sequences and OTU table. Representative sequences were aligned with mafft and used to construct a phylogeny with fasttree. Alpha-diversity metrics (Observed_species, Simpson) were estimated using the diversity plugin with samples were rarefied. Meanwhile, principal coordinates analysis (PCoA) was selected to illustrate the clustering of different samples. In this study, Pco1 and Pco2 were used to represent the β diversity of microbial communities. OTUs was given a taxonomy using the Silva v132 99% OTU reference sequences and the classify-sklearn nave Bayes taxonomy classifier in the feature-classifier plugin (*Liu et al., 2020b*).

Two-way ANOVAs ($p < 0.05$) was used to analyze the significant differences between nitrogen application frequency and earthworms on CO$_2$ emission and soil properties. One-way ANOVA with Duncan testing ($p < 0.05$) was used to evaluate the significant differences in soil properties and CO$_2$ emission among the six treatments. Linear regression analysis was used to study the relationship between soil respiration, soil property and soil microbial community under six treatments. Spearman's correlation analyses were performed to assess the relationships between soil properties, respiration, plant biomass and microbial community. Soil chemical properties data were analyzed with

SPSS software (version 26; IBM, Chicago, IL, USA). We conducted a classification random forest analysis to identify the major statistically significant microbial predictors of the composition (relative abundance: number of sequences of major phyla/class/order level) of bacteria and fungi acting on soil respiration. The analysis was conducted using the rfPermute package of the R (4.2.2; *R Core Team, 2023*) statistical software. The significant predictors from random forest analysis were further selected for structural equation modeling (SEM) analysis. SEM analysis was applied to determine the direct and indirect contributions of soil properties and the bacterial community to $CO_2$ emission. SEM analysis was performed using AMOS 22.0 software (SPSS, Chicago, IL, USA). The model fitness was evaluated by $\chi2$ ($p > 0.05$), comparative fit index, and root mean square error of approximation.

# RESULTS

## Nitrogen and earthworm application effects on soil respiration

Soil respiration varied with Soybean growth period, showing obvious seasonal variation (Fig. 1A). It was the lowest during the early vegetative stage (June 23rd–July 14th), reached maximum at the reproductive stage (July 21st–Sep. 7th), and started declining during the maturity period (Sep. 7th–Sep. 17th). Total soil respiration varied during the study from 1.13 to 4.63 μmol m$^{-2}$ s$^{-1}$, with an average of 2.55 ± 0.12 μmol m$^{-2}$ s$^{-1}$ (Fig. 1B). Compared with C, soil respiration in the NL increased significantly by 25%. E*NH, E*NL, NH, and E increased by 21%, 21%, 12%, and 11%, respectively. Soil respiration was significantly affected by nitrogen application, but not by earthworm addition or interaction between nitrogen and earthworm addition (Table 1).

## Nitrogen and earthworm application effects on soil properties and plant index

Nitrogen application significantly influenced soil pH, TN, $NO_3^-$-N, and $NH_4^+$-N. No effects of earthworm on soil properties were found. A significant interactive effect between earthworms and nitrogen application was found on grain yield (Table 1). The E*NL treatment had the highest soil TN, which was also significantly higher than the C and E treatments (Fig. 2). In comparison to the C treatments (Fig. 2), the soil $NO_3^-$-N was considerably greater in nitrogen application treatments (NH, NL, E*NH, and E*NL). Soil $NH_4^+$-N was significantly higher in NH and E*NH than in other treatments (Fig. 2). Soil pH of E*NL, NL, NH, and E*NH decreased by 3.38%, 3.03%, 2.17% and 1.64% respectively and E increased 1.26% compared with the C treatments. The highest grain yield was observed in E*NH, which was also significantly greater than NH (Fig. 2). However, neither the application of nitrogen nor its interaction with earthworms had a significant influence on soil temperature, TC, plant biomass, plant height, NPP, NGP and W100.

## Soil properties, plant index and microbial community regulated $CO_2$ emission

Soil respiration showed a positive correlation with Nitrate N ($R^2 = 0.10$, $p < 0.05$, Fig. S1), aboveground biomass ($R^2 = 0.16$, $p < 0.05$, Fig. S1), plant height ($R^2 = 0.15$, $p < 0.05$,

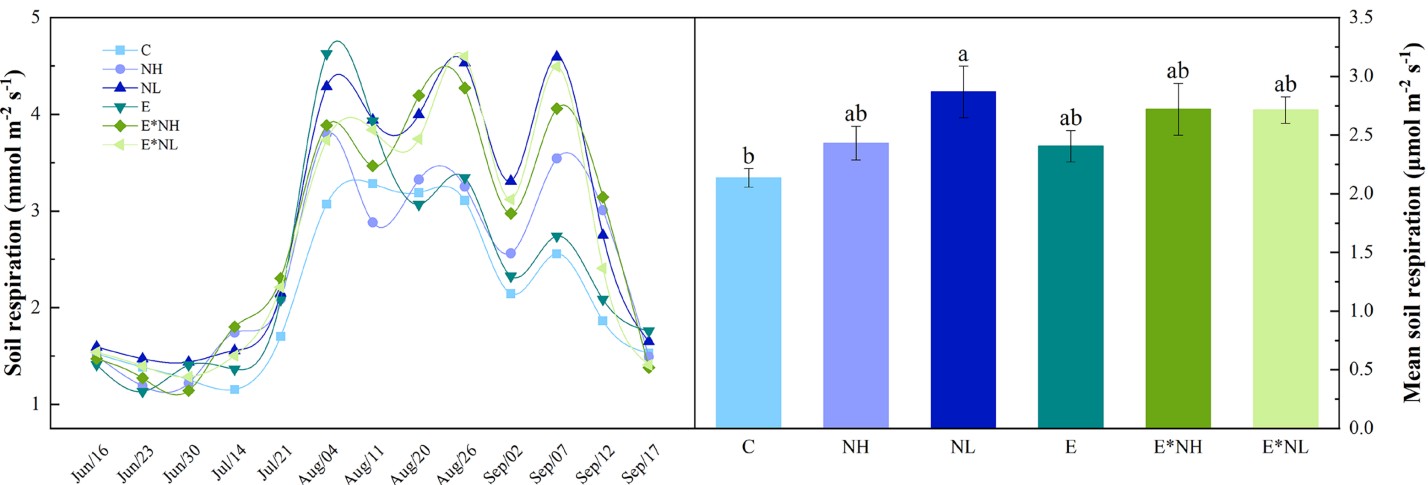

**Figure 1** **Seasonal dynamics and mean values of soil respiration under the six treatments in 2022 (M ± SE, *n* = 5).** C, Control; NH, high-frequency nitrogen application; NL, low-frequency nitrogen application; E, earthworm addition; E*NH, earthworm and high-frequency nitrogen application; E*NL, earthworm and low-frequency nitrogen application.

**Table 1** **Results (F and *p* values) of two-way ANOVAs on the effects of nitrogen application (N), earthworm addition (E), and their interactions on soil respiration (SR) and other indexes.**

| Variables | N | E | N × E |
|---|---|---|---|
| SR | 5.322* | 1.202 | 1.281 |
| ST | 0.210 | 0.034 | 0.581 |
| SM | 2.248 | 2.202 | 3.042 |
| Soil pH | 5.370* | 0.186 | 0.241 |
| TN | 4.566* | 0.031 | 3.157 |
| TC | 0.394 | 0.440 | 0.529 |
| $NO_3^--N$ | 14.581*** | 0.417 | 1.600 |
| $NH_4^+-N$ | 6.207** | 1.515 | 0.915 |
| AGB | 0.750 | 1.967 | 2.675 |
| RB | 0.301 | 0.121 | 1.516 |
| PLH | 2.137 | 0.723 | 1.563 |
| NPP | 0.270 | 0.002 | 2.326 |
| NGP | 0.355 | 0.002 | 2.897 |
| GRY | 2.838 | 2.650 | 3.651* |
| W100 | 0.893 | 2.929 | 0.907 |

**Notes:**
Statistical differences are indicated as:
* $p < 0.05$.
** $p < 0.01$.
*** $p < 0.001$.

Fig. S1), grain yield ($R^2$ = 0.12, $p < 0.05$, Fig. S1), and negatively with soil pH ($R^2$ = 0.18, $p < 0.05$, Fig. S1), but it was not correlated with root biomass (Fig. S1).

The random forest modeling showed the major bacterial and fungal phyla, classes, and orders for predicting soil respiration. These taxa include several bacterial and fungal such as Corynebacteriales, Myxococcales, Sordariomycetes, Verrucomicrobia, Thermoleophilia,

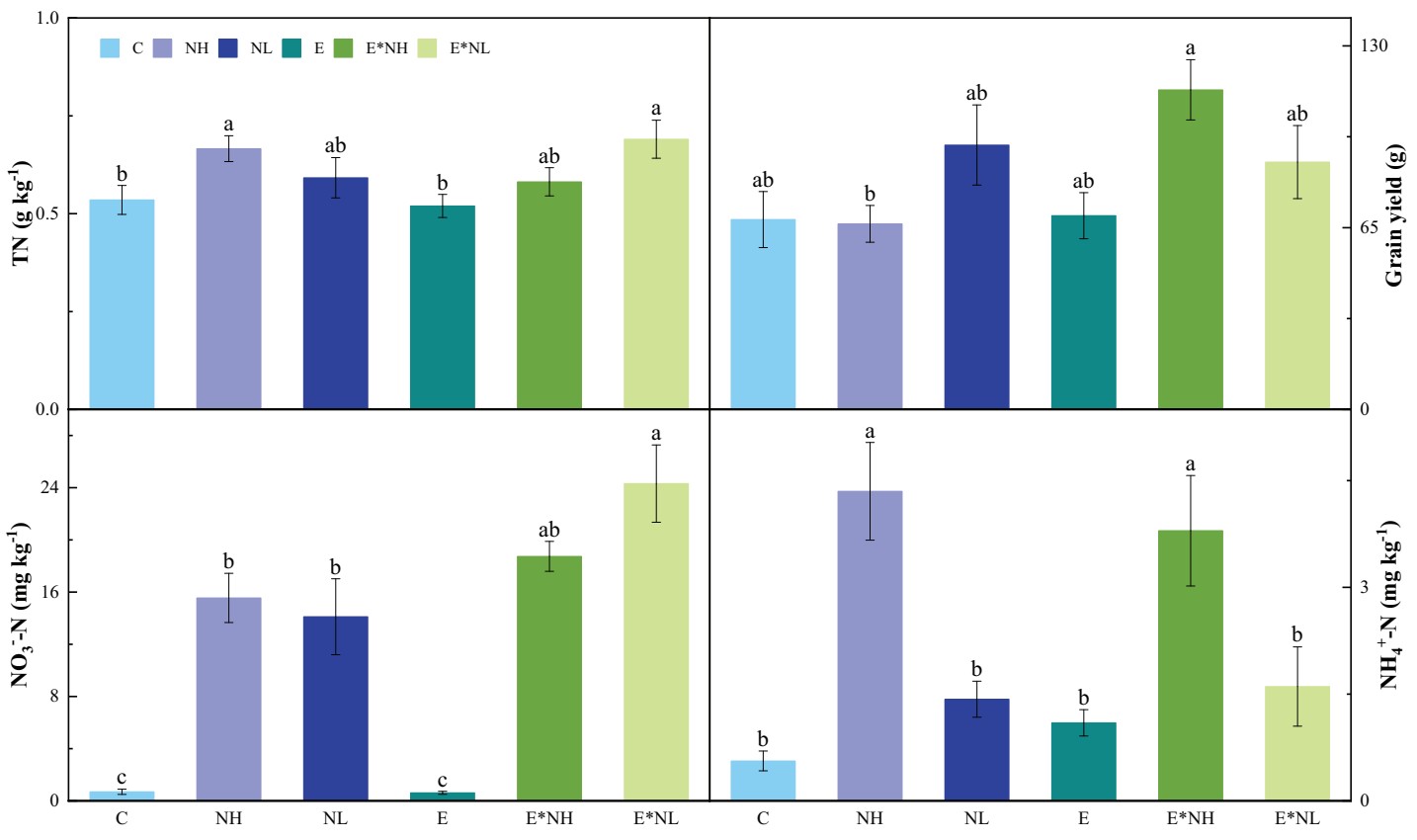

**Figure 2** Effects of nitrogen addition (NH, NL) and earthworm addition (E) on total N (TN), grain yield (GRY), nitrate N (NO$_3^-$-N) and ammonium N (NH$_4^+$-N) in 2022 (M ± SE, $n$ = 5).

Agaricomycetes, Gammaproteobacteria, Blastocladiomycetes, and Proteobacteria (Fig. 3). In addition, soil respiration was significantly positively correlated with the relative abundance of Gammaproteobacteria (R$^2$ = 0.149, $p$ < 0.05, Fig. S2) and Corynebacteriales (R$^2$ = 0.154, $p$ < 0.05, Fig. S2), as well as bacteria Pco1 (R$^2$ = 0.109, $p$ < 0.05, Fig. S2), and negatively correlated with the relative abundance of Myxococcales (R$^2$ = 0.161, $p$ < 0.05, Fig. S2), Verrucomicrobia (R$^2$ = 0.122, $p$ < 0.05, Fig. S2) and Thermoleophilia (R$^2$ = 0.115, $p$ < 0.05, Fig. S2). The Sordariomycetes, Agaricomycetes, Blastocladiomycetes and Proteobacteria were not correlated with soil respiration ($p$ > 0.05). Notably, the relative abundance of Thermoleophilia, Myxococcales and Verrucomicrobia was low in NL, while the relative abundance of Corynebacteriales and Gammaproteobacteria was high. In general, NL treatment raised the relative abundance of copiotrophs while decreasing the relative abundance of oligotrophs.

The Spearman's correlation coefficients between the microbial characteristics and the soil respiration as well as soil properties were estimated (Table 2). Soil respiration rate was negatively correlated to bacterial Simpson. In particular, there was a correlation between the respiration and the relative abundance of major bacterial and fungal phyla. The strong correlation between soil respiration and the abundances of Blastocladiomycetes and Agaricomycetes in the fungal compositions, as well as Gammaproteobacteria,

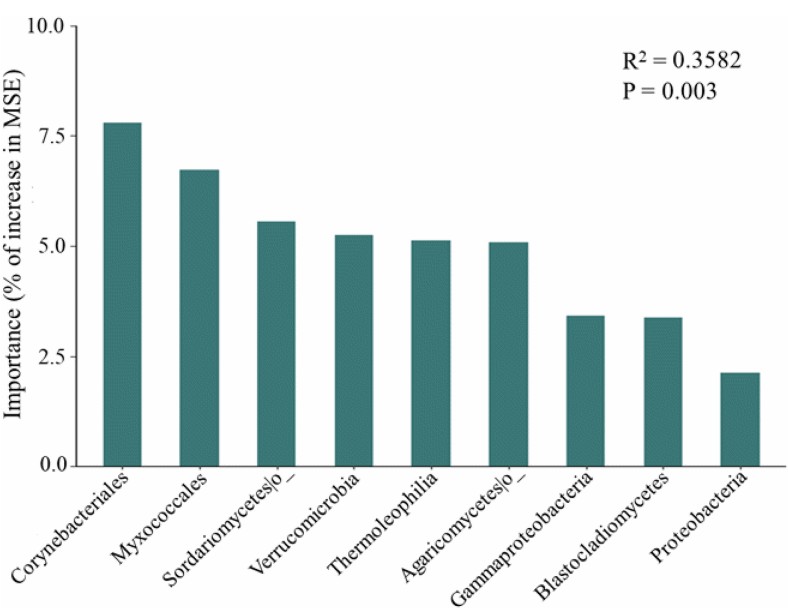

**Figure 3 Predictor importance of major bacterial and fungal phyla/classes as drivers of soil respiration based on random forest model.**

**Table 2 Spearman's rank correlation coefficients (ρ) between the microbial (bacterial and fungal) characteristics soil properties, and plant biomass as well as soil respiration.**

| Microbial community | SR | ST | SM | SWC | Soil pH | TN | TC | $NO_3^-$-N | $NH_4^+$-N | AGB | RB |
|---|---|---|---|---|---|---|---|---|---|---|---|
| Bacterial Richness | −0.022 | −0.226 | −0.273 | −0.192 | −0.044 | 0.153 | −0.265 | −0.390* | −0.113 | −0.078 | −0.154 |
| Bacterial Simpson | −0.459* | −0.109 | 0.026 | −0.001 | 0.096 | 0.061 | 0.096 | −0.183 | −0.254 | −0.346 | −0.486** |
| Bacterial Pco1 | 0.281 | −0.658** | 0.054 | 0.239 | −0.320 | 0.325 | 0.640** | 0.080 | −0.093 | 0.380* | 0.285 |
| Bacterial Pco2 | 0.204 | 0.393* | 0.305 | 0.051 | −0.411* | 0.091 | −0.083 | 0.706** | 0.172 | 0.167 | −0.193 |
| Fungal Richness | −0.146 | 0.020 | −0.166 | −0.357 | 0.383* | −0.161 | −0.324 | −0.340 | −0.048 | −0.087 | −0.010 |
| Fungal Simpson | −0.315 | −0.104 | 0.216 | 0.013 | 0.024 | 0.189 | 0.256 | −0.019 | −0.010 | −0.214 | −0.402* |
| Fungal Pco1 | 0.250 | −0.232 | −0.299 | −0.066 | −0.081 | 0.022 | −0.096 | −0.212 | 0.061 | 0.293 | 0.503** |
| Fungal Pco2 | 0.145 | −0.303 | 0.349 | 0.063 | −0.326 | 0.206 | 0.434* | 0.051 | −0.003 | 0.129 | −0.064 |
| Proteobacteria | 0.276 | 0.016 | 0.100 | 0.304 | −0.176 | 0.123 | 0.254 | 0.373* | 0.135 | 0.142 | −0.119 |
| Verrucomicrobia | −0.439* | 0.078 | 0.068 | −0.133 | 0.134 | −0.001 | −0.063 | −0.196 | 0.136 | −0.217 | −0.208 |
| Gammaproteobacteria | 0.380* | 0.148 | 0.224 | 0.044 | −0.221 | 0.165 | 0.087 | 0.479** | 0.199 | 0.156 | −0.076 |
| Thermoleophilia | −0.399* | 0.530** | −0.051 | −0.562** | 0.500** | −0.362* | −0.565** | −0.306 | 0.016 | −0.402* | −0.194 |
| Myxococcales | −0.483** | −0.020 | −0.238 | 0.202 | 0.366* | −0.184 | 0.033 | −0.380* | −0.277 | −0.413* | −0.330 |
| Corynebacteriales | 0.304 | −0.252 | −0.113 | 0.177 | −0.044 | 0.011 | 0.167 | −0.131 | 0.041 | 0.077 | 0.421* |
| Blastocladiomycetes | 0.421* | −0.056 | 0.227 | 0.165 | −0.297 | 0.466** | 0.067 | 0.481** | 0.274 | 0.234 | 0.013 |
| Sordariomycetes_o_ | −0.047 | 0.075 | −0.006 | −0.237 | −0.135 | 0.089 | −0.090 | −0.133 | −0.007 | 0.162 | −0.120 |
| Agaricomycetes_o_ | −0.427* | −0.017 | −0.094 | 0.272 | 0.101 | 0.048 | 0.009 | 0.019 | 0.003 | −0.462* | −0.225 |

**Notes:**
** Correlation is significant at the 0.01 level.
* Correlation is significant at the 0.05 level.
Number of OTUs, richness; alpha diversity, Simpson; beta diversity, Pco1, Pco2; SR, soil respiration; ST, soil temperature; SM, soil moisture; Soil pH; TC, total C; TN, total N; $NO_3^-$-N, nitrate N; $NH_4^+$-N, ammonium N; AGB, aboveground biomass; RB, root biomass.

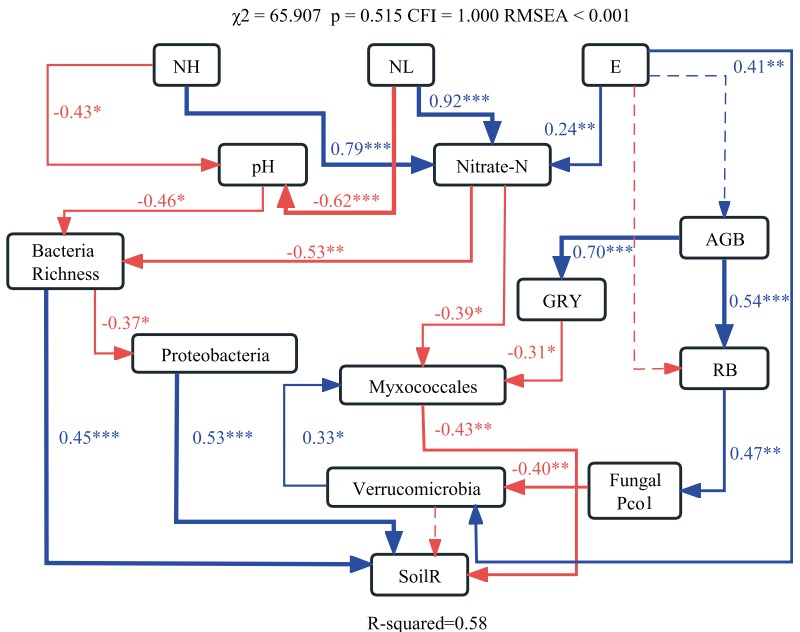

$\chi2 = 65.907$  $p = 0.515$ CFI = 1.000 RMSEA < 0.001

**Figure 4 Structural equation modeling (SEM) showing effects of soil abiotic and biotic properties on soil respiration.** Blue and red lines indicate significant positive and negative relationships, respectively. ***$p$ < 0.001; **$p$ < 0.01; *$p$ < 0.05.

Verrucomicrobia, Thermoleophilia, and Myxococcales in the bacterial compositions was found (Table 2).

The optimum structural equation model (SEM) revealed that the interactive networks of abiotic and biotic factors regulated soil respiration ($\chi^2$ = 65.907, $p$ = 0.515, CFI = 1.000, RMSEA < 0.001). Results of SEM explained 58% of the variations in soil respiration (Fig. 4). Changes in pH and $NO_3^-$-N caused by nitrogen application altered bacterial richness and the relative abundance of Myxococcales, which affected soil respiration.

# DISCUSSION

## Effects of nitrogen application frequency on soil respiration and microbial community

Nitrogen application increases crop yield and quality (*He et al., 2022*). But fertilization practices can significantly impact on soil $CO_2$ emissions (*Wang et al., 2021*). In the study, fertilizer treatments considerably improved soil $CO_2$ emission by 12–25% in comparison to the control treatment, which is consistent with previous study findings (*Yan et al., 2021*; *Lamptey et al., 2019*). The frequency of nitrogen application (two and twelve times) had different pulse effects and influenced response variables such as soil nitrogen, bacterial diversity and root foraging requirements, which demonstrated the importance of nitrogen application frequency on nitrogen deposition in ecosystems (*Cao et al., 2019*). The NL resulted in lower pH (*Ning et al., 2015*; *Ning et al., 2022*) and $NH_4^+$ (*Zhou et al., 2016*), but greater acidity than that the NH. The reason is that the $H^+$ generated by nitrification reacted with the salt ions adsorbed on the colloidal surface of the soil, and the $H^+$ was adsorbed on the surface of the soil, resulting in a decrease in soil pH and soil acidification

(*Sun et al., 2006*). Moreover, the single nitrogen application of NL was higher than that of NH, which was difficult for plants to absorb in a short time. More nitrogen is used for microbial nitrification, so more $NH_4^+$ is available for nitrification, and ultimately more $H^+$ is produced (*Zhou et al., 2016*). Lower soil pH due to nitrogen application led to increases in soil respiration through changing the bacterial community (*He et al., 2022*; *Whitaker et al., 2014*; *Liu et al., 2020a*). In addition, high-frequency applications resulted in stronger levels of soil N content. Soil $NO_3^-$-N is higher in the NH than the NL treatments, the reason is that higher $NH_4^+$ exacerbates the toxicity of alkali ions inhibiting growth and $NO_3^-$-N uptake of root (*Hao et al., 2022*; *Ning et al., 2022*). Our study revealed that $NO_3^-$-N and soil pH were important environmental factors explaining changes in bacterial community composition, which is consistent with previous studies (*Chen et al., 2021a*).

Most bacteria are sensitive to pH variations and prefer being neutral (*Anil, Menka & Preeti, 2019*). Moreover, the bacterial communities control the major of ecological activities in agricultural ecosystems (*Van der Heijen, 2008*). In our study, acidification of alkaline soil made an ideal habitat for the soil microbial community. $NO_3^-$-N is a substrate that provides nutrients to microorganisms and is associated with increased richness of soil bacterial communities, which is consistent with previous studies (*Chen et al., 2021a*). Consequently, the reduced pH and $NH_4^+$ toxicity resulted in the highest bacterial richness in the NL, which directly enhanced soil respiration. In this study, NL treatments increased the bacterial richness, whereas decreased the fungi proportion. Higher bacterial richness was consistent with enhanced soil respiration, suggesting that increased metabolic rates drive this reaction (*Hagerty et al., 2014*).

## Effects of earthworm on soil respiration and microbial community

Earthworm activity can increase the contents of soil active organic carbon, soil inorganic nitrogen, microbial biomass carbon, and microbial biomass nitrogen (*Yu et al., 2007*; *Li et al., 2022*). Moreover, earthworm addition may also increase soil $CO_2$ emissions (*Lubbers et al., 2013*). In this study, earthworm addition enhanced soil $CO_2$ emissions, which is consistent with previous research results (*Lubbers et al., 2013*; *Song et al., 2020*; *Yang et al., 2019*). Earthworm addition did not significantly increase soil respiration in our study, which is attributed to the fact that low density addition and type of earthworm were endophytic. Endophytic earthworms are mainly active in deeper soil layers, which have the characteristics of slow sexual maturity, limited population and weak respiration (*Zhang et al., 2011*).

The increase in pH and $NH_4^+$-N caused by the addition of earthworms was consistent with previous studies (*Wang et al., 2013*; *Ferlian et al., 2020*; *Na et al., 2023*). Earthworm activity increases soil pH, which is related to the large amount of mucus secreted by earthworm epidermis and the amino acids, sugars and inorganic salts in earthworm gut excreta (*Ferlian et al., 2020*; *Na et al., 2023*). In the study, earthworm addition affected soil respiration by altering bacterial community composition. It has been shown that soluble carbon and other compounds released in the digestive tract of earthworms contribute to bacterial proliferation (*Barbosa et al., 2017*). Changes in pH and available nitrogen after the application of earthworm casts altered microbial community composition (*Zhao et al.,*

*2016*), which is consistent with the changes in $NO_3^-$-N caused by earthworm addition altering the relative abundance of Myxococcales and affecting soil respiration in our study.

## Effects of nitrogen application frequency and earthworm on soil respiration

In the study, the addition of earthworms and the interaction between earthworms and nitrogen had no significant effect on soil respiration, which may be because the reduction of respiration rate caused by the absorption of sufficient nitrogen by plants and the increase of respiration rate caused by earthworm addition offset each other (*Yang et al., 2019*). Another possible explanation is that earthworm addition is not adapted to the habitat and cannot survive for a long time, therefore it has a short-term effect on soil respiration (*Frouz et al., 2014*). The interaction between earthworms and high-frequency N application showed an additive effect, whereas the interaction between earthworms and low-frequency N application reduced the effect of low-frequency N application on soil respiration. This suggests that the presence of earthworms buffers the pulse of low-frequency nitrogen application.

## Microbial community regulated CO$_2$ emission

Important microbial classification and functional properties have been reported to potentially predict changes in soil respiration. Based on the study of the random forest model, we were able to pinpoint the primary microbial taxa that predict soil respiration. We also found that the dominant bacteria in the microbial community explained 35.82% of the variation in soil respiration (Fig. 3). Dominant bacteria of Myxococcales have been shown to be key factors in regulating soil respiration in our study, which is consistent with a previous study (*Liu et al., 2021*). Proteobacteria have been found to prefer soils with abundant carbon availability, hence this phylum might encourage increases in respiration. In addition, soil respiration was substantially connected with Gammaproteobacteria, Sordariomycetes, and Agaricomycetes (*Ren et al., 2018*), and Verrucomicrobia (*Han & Wang, 2023*), which is consistent with predicted species by random forest model in our study. The Sordariomycetes, Agaricomycetes, Blastocladiomycetes and Proteobacteria did not correlate with soil respiration in our study, which may be attributed to competition among microorganisms and the importance of dominant species (*Ren et al., 2018*). According to generally believed accounts, copiotrophs and oligotrophs have ability to use C for respiration. In general, it has been suggested that oligotrophs have slower respiration rates than copiotrophs (*Liu et al., 2018*; *Chen et al., 2021b*; *Liu et al., 2020b*). Myxococcales and Gammaproteobacteria are considered potential copiotrophs, but Verrucomicrobia, Thermoleophilia, and Corynebacteriales are classified as oligotrophs (*Chen et al., 2021b*). Moreover, the relative abundance of oligotrophs (Thermoleophilia and Corynebacteriales) is lowest in the low-frequency nitrogen application treatments, which can help explain the highest soil respiration in the NL plots. Importantly, our revealed that soil properties like pH and $NO_3^-$-N affected soil respiration rates indirectly by changing the soil microbial community rather than directly (*Liu et al., 2021*; *Wang et al., 2022*). Our results highlight

the importance of keystone taxa and microbial community composition as predictors in the development of the soil carbon model.

## CONCLUSIONS

This experimental study demonstrates that low frequency nitrogen application has a significant impact on $CO_2$ emissions. Our results suggest that the rate increased soil respiration by changes in soil properties, which altered bacterial richness and keystone taxa. Furthermore, these findings not only highlight the importance of bacterial community composition and keystone taxa regulating soil $CO_2$ emissions, but provide data support and fill the knowledge gap for studying the effects of nitrogen application frequency and soil animals on $CO_2$ emissions in the Huang-Huai-Hai Plain.

## ACKNOWLEDGEMENTS

We are grateful to the many graduate students, field and lab assistants who helped with data collection and analyses since 2022.

### Funding

This work was supported by the National Natural Science Foundation of China (42107225; 31770522; 32130066). The funders had no role in study design, data collection and analysis, decision to publish, or preparation of the manuscript.

### Grant Disclosures

The following grant information was disclosed by the authors:
National Natural Science Foundation of China: 42107225, 31770522, 32130066.

### Competing Interests

The authors declare that they have no competing interests.

### Author Contributions

- Mei Guang Jiang conceived and designed the experiments, performed the experiments, analyzed the data, prepared figures and/or tables, authored or reviewed drafts of the article, and approved the final draft.
- Jingyuan Yang performed the experiments, authored or reviewed drafts of the article, and approved the final draft.
- Qi Xu performed the experiments, authored or reviewed drafts of the article, and approved the final draft.
- Linyu Qi performed the experiments, authored or reviewed drafts of the article, and approved the final draft.
- Yue Gao performed the experiments, authored or reviewed drafts of the article, and approved the final draft.
- Cancan Zhao conceived and designed the experiments, authored or reviewed drafts of the article, and approved the final draft.

- Huijie Lu performed the experiments, authored or reviewed drafts of the article, and approved the final draft.
- Yuan Miao conceived and designed the experiments, authored or reviewed drafts of the article, and approved the final draft.
- Shijie Han conceived and designed the experiments, authored or reviewed drafts of the article, and approved the final draft.

## DNA Deposition

The following information was supplied regarding the deposition of DNA sequences:

The sequences are available at NCBI (PRJNA1008076) and figshare: Miao, Yuan (2023). Supplement Sequence Data-PeerJ.zip. figshare. Dataset. https://doi.org/10.6084/m9.figshare.23937741.v1.

## Data Availability

The raw measurements are available in the Supplemental Files.

## Supplemental Information

Supplemental information for this article can be found online at http://dx.doi.org/10.7717/peerj.17176#supplemental-information.

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
