# Peer review of "The responses of CO2 emission to nitrogen application and earthworm addition in the soybean cropland"

_PeerJ, doi:10.7717/peerj.17176_

## Round 0.1 · original submission · Major Revisions

Dear Dr. Miao,

We received two evaluations for your paper. While one reviewer recommends rejection, the second is slightly more positive. This means that, in order to be accepted, the paper requires additional work.

I recommend also that you clearly state the limitations present in this study. I add that, in my opinion, you introduced too many variables which can be hardly analysed in a single paper. For instance, the addition of earthworms is not essential and could be presented in a separate paper.

Please take your time for the revision and consider carefully all the comments, although you are not obliged to follow them all.

Sincerely,

Leonardo Montagnani

·

Basic reporting

The article uses clear English and sufficient background citations to show the work fits into the field of knowledge. The figures did not use high dpi format which influence the resolution. The sequences data did not upload to NCBI or any public website.

Experimental design

The experimental design is acceptable, but some detail information are missing and dis-matched in different places of the manuscript.

Validity of the findings

no comment

Additional comments

The manuscript titled “Low-frequency nitrogen deposition addition affects CO2 emission through regulating soil bacterial community composition” studied the effect of N addition with different frequency on the soil respiration and bacterial communities. The topic is interesting but the manuscript has several dis-match and mistakes which need to be improved a lot before it can get published. The discussion did not give a profound explanation for the results, in most case, only compare the result with others. See below detail comments:
Abstract:
1.The first sentence and the results showed earthwarms, but no earthwarms shown in the title.
2.Line 24, N not necessary to usecapital letter
3.Line 29: community composition, ......fungal diversity? This study focus bacteria
Line 114: how about the unit of earthwarms added? per ha?
Line 118: what is the number mean?
Line 134: three soil cores, but in line 113, the authors said 5 replicates,
Line 173: there should be a space in front of briefly
Line 254: check AGB, GRY, RB and other short name, give the full name for the first appearance
Line 383-385:has logic error.
References list:
Checkout the whole list carefully, many mistakes
Line 405: latin name should be use italic stytle
Line 410: forest
Line 429: different format
Line 464: capital letters for the title, different style
Line 595: the authors seem missing

Reviewer 2 ·

Basic reporting

1 The writing needs a lot of work to be published. I have the ambiguous sentences highlighted in the manuscript.

2 The title is "Low-frequency nitrogen deposition addition affects CO2 emission through regulating soil bacterial community composition". However, the authors investigated effects of different N deposition frequency, earthworm, and their interaction on soil CO2 emission and microbial communities. The title and the content did not match well. It is necessary to revise the title to highlight the earthworm, and the ecosystem.

3 The figures are not clear.

4 Most importantly, did the authors tried to evaluate the effects of N deposition or N application? The two terms are different which decide whether the experiment design in the present study makes sense. If it is N deposition, how authors decide the frequency and deposition rates? The authors need to make it clear.

Experimental design

1 The authors aimed to investigate effects of different N deposition frequency on soil respiration and microbial community. They compared continuous N addition for twelve months and twice N addition both at the same total amount. It is true that few research did this experiment. However, the authors did not judge their design very well, e.g., how they came out the total amount? Did twelve months and twice per year mimic the natural N deposition? In reality, when will the two situations happen? Same issue is for the earthworm addition. What is the amount of earthworm individuals you added? based on what density in the field?

Validity of the findings

The experiment had five replicates for each treatment.
Fig. 4 and Fig. 5 may be incorporated into Table 2.
The conclusion needs to be more focused. Too much was mentioned.

Additional comments

na

Annotated reviews are not available for download in order to protect the identity of reviewers who chose to remain anonymous.

---

## Round 0.2 · Major Revisions

Dear Dr, Miao,

We received an additional evaluation of your work. The reviewer expressed his/her doubts about your work, which I largely share.

In addition, I have concerns about the experimental setup: were the replicates enough? how do you explain that fewer applications of fertilizer led to higher nitrogen content? What was the nitrogen content at the different plots before and after the treatment?

I also have doubts about the effect of earthwarms application: how is possible that living organisms do not enhance soil respiration?

Overall, I recommend that you do not confound correlation with causality, as it seems you are doing in the abstract.

Please consider if you can manage all these problems in a deeply revised version of the article, or if is it better to consider another, less demanding, journal.

Sincerely,

Leonardo Montagnani

Reviewer 2 ·

Basic reporting

The English still needs more work. I marked some in the attached manuscript. Strongly suggest the English to be professionally polished.

Experimental design

The authors added more details about the experimental design of earthworm and N fertilizer frequency. However, is splitting N fertilization for soybean into 12 times in reality? Based on my knowledge, for soybean, no N fertilization or one-time fertilization is enough.

The authors only measured microbial community at harvest time and mentioned using microbial community to predict soil respiration. However, microbial community is very dynamic with time. So is it meaningful and practical to use microbial community at harvest to predict soil respiration during the growing period? I would think it is more reasonable to discuss effects of N application and earthworm on soil respiration and microbial community, instead using microbial community to predict soil respiration. Soil properties may be better predictors for soil respiration as mostly discussed.

Validity of the findings

Using the microbial community to predict soil CO2 emission seems novel but I am not sure it is appliable.

Additional comments

Line 122-123: Why earthworms were added at this time?
Line 135: “In September” means at harvest? Need to be more specific.
Line 213-214: mmol or nmol? Please check.
Line 266 “Order to increase crop output and soil quality,...”? “In order to increase crop yield and improve soil quality?”?
Line 270-272 “In addition, NL promoted soil respiration more than NH, indicating that nitrogen inhibited soil respiration with the increase of nitrogen frequency.” Are you sure?
Line 266-283: I don’t think this paragraph justified the authors’ idea very well.
Line 288-289: “As far as we know, there is almost no research on the impact of nitrogen application at different frequencies on bacterial richness.” Do you know why?

Fig. 6 There are no abcdef in the figure.

Annotated reviews are not available for download in order to protect the identity of reviewers who chose to remain anonymous.

---

## Round 0.3 · Minor Revisions

Dear Dr Miao,

We received an additional evaluation of your work, suggesting a minor revision.

I also considered your paper personally. While I find the answers adequate, I still find some parts of the text unclear.

I have these additional suggestions:

Please check carefully that all the necessary information is presented in the methods section to make the experiment replicable. This applies also to the observation that the considered earthworms are endophytic

Please ask a professional editor to make your text more fluent and grammatically correct.

Please check that all, or at least the most relevant information present in the answers is also in the text.

Please remember that correlation does not mean causality. Therefore please state that the different bacterial species correlate (or not) with soil respiration.

Please check the value of R2 reported in Figure 3: to my understanding, it should be between 0 and 1.

Sincerely,

Leonardo Montagnani

Reviewer 2 ·

Basic reporting

The current form has been improved compared to the last one. The writing still has room to improve.
line 266-268: I did not get what you tried to say.
line 268-269: Though I know what 2N and 12N mean, the authors didn't mention it before.
line 269-270: This only explains why nitrogen application decreased pH but not why 2N applications caused greater acidity.
line 330: soil respiration

Experimental design

The experiment design is clear now.
Did the authors measure soil DOC which is a important indicator of soil respiration?

Validity of the findings

I am just curious that for the earthworm addition, the increased CO2 emission is not by the earthworm itself?

Additional comments

na

---

## Round 0.4 · accepted · Accept

Dear Dr. Miao,

I am pleased to inform you that I consider your paper acceptable now.

Sincerely,

Leonardo Montagnani